# Exploring the Heart–Mind Connection: Unraveling the Shared Pathways between Depression and Cardiovascular Diseases

**DOI:** 10.3390/biomedicines11071903

**Published:** 2023-07-05

**Authors:** Justyna Sobolewska-Nowak, Katarzyna Wachowska, Artur Nowak, Agata Orzechowska, Agata Szulc, Olga Płaza, Piotr Gałecki

**Affiliations:** 1Department of Adult Psychiatry, Medical Univeristy of Lodz, 90-419 Lodz, Poland; katarzyna.wachowska@umed.lodz.pl (K.W.); agata.orzechowska@umed.lodz.pl (A.O.); piotr.galecki@umed.lodz.pl (P.G.); 2Department of Immunopathology, Medical Univeristy of Lodz, 90-419 Lodz, Poland; artur.nowak@stud.umed.lodz.pl; 3Psychiatric Clinic of the Faculty of Health Sciences, Medical University of Warsaw, 02-091 Warsaw, Poland; agata.szulc@wum.edu.pl (A.S.); olga.w.plaza@gmail.com (O.P.)

**Keywords:** depression, cardiovascular disease, obesity, diabetes, civilization diseases, mental disorders, risk factors, inflammation, interleukins, comorbidity

## Abstract

Civilization diseases are defined as non-communicable diseases that affect a large part of the population. Examples of such diseases are depression and cardiovascular disease. Importantly, the World Health Organization warns against an increase in both of these. This narrative review aims to summarize the available information on measurable risk factors for CVD and depression based on the existing literature. The paper reviews the epidemiology and main risk factors for the coexistence of depression and cardiovascular disease. The authors emphasize that there is evidence of a link between depression and cardiovascular disease. Here, we highlight common risk factors for depression and cardiovascular disease, including obesity, diabetes, and physical inactivity, as well as the importance of the prevention and treatment of CVD in preventing depression and other mental disorders. Conversely, effective treatment of CVD can also help prevent depression and improve mental health outcomes. It seems advisable to introduce screening tests for depression in patients treated for cardiac reasons. Importantly, in patients treated for mood disorders, it is worth controlling CVD risk factors, for example, by checking blood pressure and pulse during routine visits. It is also worth paying attention to the mental condition of patients with CVD. This study underlines the importance of interdisciplinary co-operation.

## 1. Introduction

Depression and cardiovascular disease are both serious, growing problems in the modern world. They belong to group of diseases known as civilization diseases, meaning that they affect many people worldwide. As stated by the WHO, depression affects around 5% of adult people, an estimated 280 million people. Depression might lead to suicide and 700,000 people die by suicide every year [1]. When it comes to cardiovascular disease, it has been estimated that 17.9 million people died from CVDs in 2019, which account for 32% of all global deaths. Of these deaths, 85% were due to heart attack and stroke [2]. It is, therefore, easy to understand that both of these diseases are strongly associated with not only severe impairment in everyday functioning but also the risk of premature death. Depressive disorder has been generally associated with CVD [3] and it has been observed that depression with atypical features is linked even more strongly with cardiac risk [4]. Depression has been observed as an independent risk factor for cardiac health problems [5] as well as a consequence of CVD [6]. It was also associated with the morbidity and mortality of cardiovascular disease [7]. When gender is taken into account, it has been observed that depression causes a greater increase in CVD incidence in women, and females suffering from CVD experience higher levels of depression than men [6]. Although well-observed associations between depression and CVD have been described, it has not been clearly established how these disorders interact yet. Possible risk factors combining these two very different states have been described and involve, among others, inflammation [8]. The presented article explores common risk factors between these two seemingly very different states, but starts off with basic information about the discussed disorders.

Civilization diseases are defined as non-communicable diseases that affect a large proportion of the population. They are considered to pose a serious, still-growing problem that affect people in low- and middle-income countries disproportionately strongly, where more than three quarters of global NCD deaths (31.4 million) occur. We can point out examples of conditions such as obesity, diabetes, cardiovascular disease, cancer, Alzheimer’s disease, and depression [9,10].

The frequent occurrence of these diseases is characteristic of highly developed areas in terms of economy and civilization. Anti-health behaviors and other risk factors among citizens of wealthy countries are the main causes of a poor health condition [11]. It should be remembered that these diseases significantly reduce the quality of life and, in the long run, can lead to death.

Depression is a common chronic mental disorder that affects thoughts, mood and physical health [12]. To be diagnosed with a major depressive episode according to the DSM-V (Diagnostic and Statistical Manual of Mental Disorders) classification, a patient should have at least five of the symptoms presented in Table 1 that have been present over the same 2-week period and which indicate a change from previous functioning:

Please note that an episode cannot be attributed to the physiological effects of a substance or to any other medical condition. The occurrence of a major depressive episode is not better explained by the presence of schizoaffective disorder, schizophrenia, schizophrenia-like disorder, delusional disorder or other schizophrenia spectrum disorder, and other psychotic disorders. There also must be no history of a manic or hypomanic episode [13].

As we can see, the symptoms of depressive disorders vary from person to person. For many years, researchers have mainly focused on the causes of mental illness, including mood disorders. In the late 1990s, a link between a reduced brain activity and reduced cortical volume in patients suffering from bipolar and unipolar depression was discovered [14]. Later, histopathological examination of the brains of patients with mood disorders showed a lower cortical thickness and cell density in the prefrontal cortex [15]. Volumetric magnetic resonance imaging studies of patients with major depressive disorders compared to healthy controls suggested a complete reduction in brain volume, especially in the anterior cingulate and orbitofrontal cortex [16]. Additionally, of interest are the results of a meta-analysis performed by Hamilton et al. who observed significant differences in amygdala volume between treated and untreated depressed patients [17]. Ménard et al., in their paper, summarized the findings and insights for which parallel results were obtained in people with depression and models of mood disorders in rodents to investigate the potential etiology of depression. It has been discovered that several factors may be responsible for the development of depressive disorders: immunological, immune and inflammatory factors, psychological factors (e.g., stressful life events, childhood and adolescence), and neurobiological factors (participation of astrocytes in neurovascular coupling and neuronal functioning) [18].

All these discoveries can help us understand the so-called neuropathology of depressive syndrome. However, the complexity of the neurobiological processes involved in the disease make depression a conundrum of a syndrome and raise important questions regarding the importance of different brain regions (e.g., limbic structures) in the neuropathology of the disease [19]. It is a global health problem, as it is estimated that up to 5.0% of adults suffer from depression [20]. Among people suffering from depression, 40–80% of patients have suicidal thoughts, 20–60% make suicide attempts, and as many as 15% of patients successfully take their own lives. Every year, one million people worldwide die of depression, and 3800 people die from it every day [21]. According to statistics published in 2021 by the World Health Organization, more than 700,000 people die by suicide each year. This problem also affects young people; suicide is the fourth-most common cause of death at the age of 15–29 [1].

Cardiovascular disease (CVD) is the collection of diseases of the heart and circulatory system. It is a group of heterogeneous diseases sharing a common main cause of development, namely atherosclerosis. CVDs are chronic diseases that develop gradually throughout life and are asymptomatic for a long time [22]. The first group of causes includes ischemic heart disease, coronary artery disease (e.g., myocardial infarction), cerebrovascular diseases (e.g., stroke), and aortic and arterial diseases, including hypertension and peripheral vascular diseases. Other causes of CVD include congenital heart defects, rheumatic heart diseases, cardiomyopathies, and arrhythmias [23]. Cardiovascular diseases are the most common cause of death in both developed and underdeveloped countries [20]. According to statistics published in 2021 by the World Health Organization, an estimated 17.9 million people died from cardiovascular disease in 2019, accounting for 32% of all deaths worldwide. Of these deaths, 85% were due to heart attack and stroke. Of the 17 million premature deaths (under the age of 70) due to NCDs in 2019, 38% were due to CVD [7]. As risk factors, we recognize personal characteristics or a set of these, situations, conditions and potentially life events as well as diseases occurring in the patient, which have an impact on the emergence of further health problems. CVD risk factors can be divided into three large groups: biochemical, lifestyle-related, and those related to individual predispositions [2]. In the same way, factors affecting the development of depression can be classified.

Although it is well known that depression and CVD are common comorbidities, their interactions are not fully understood. Such co-morbidity brings not only increased suffering among patients but also generates high treatment costs. Understanding this relationship seems very important given the growing problem of both depression and CVD.

This review is an attempt to answer the question of whether the following guidelines can help in coping with everyday practice or whether a different approach is needed for specific patients. It also explores ways to prevent these diseases. The authors attempt to answer the questions in Table 2.

## 2. Depression and Cardiovascular Disease

Since depression and CVD symptoms often co-occur, we conducted a literature review to determine the prevalence of both conditions and the impact of comorbidity on diagnosis, clinical outcomes, and treatment. This review aims to summarize the available information on measurable risk factors for CVD and depression based on the existing literature. The authors of this paper wish to draw clinicians’ attention to the problems of patients suffering from or at risk of comorbidities. The objectives of studying the links between depression and cardiovascular disease (CVD) can be summarized as follows. Understanding the association: The first objective is to investigate and understand the relationship between depression and CVD. This involves determining the prevalence of depression among individuals with CVD and identifying potential risk factors that contribute to the development of both conditions. Additionally, it aims to explore the impact of CVD on the onset or worsening of depression.

Identifying common mechanisms: The second objective is to uncover the shared biological, physiological, and behavioral pathways that connect depression and CVD. This includes examining factors such as inflammation, oxidative stress, hormonal imbalances, and lifestyle factors such as diet, exercise, and smoking. Understanding these mechanisms can provide insights into the underlying processes and help develop targeted interventions.

Improving risk assessment and screening: The third objective aims to enhance the identification and assessment of individuals at risk for both depression and CVD. It involves developing and validating screening tools that can identify individuals who may be susceptible to developing both conditions. Incorporating depression as a risk factor for CVD seeks to improve risk-assessment models and identify subgroups that may require specific interventions.

Enhancing treatment and prevention strategies: This objective focuses on improving treatment and prevention approaches for individuals with comorbid depression and CVD. It involves evaluating the efficacy of interventions that target both conditions and developing integrated care models that address the complex interplay between mental and physical health. Furthermore, it aims to explore the impact of treating depression on CVD outcomes, and vice versa, as well as the effectiveness of lifestyle modifications and psychosocial interventions in reducing the risk and progression of both conditions.

Informing Public Health Policies and Practices: The final objective is to inform public health policies and practices regarding depression and CVD. This involves providing evidence-based recommendations for prevention and management strategies. It also seeks to influence healthcare policies to prioritize integrated care for individuals with comorbid depression and CVD, promote mental health awareness, reduce stigma, and foster collaboration between mental health and cardiovascular care providers.

By pursuing these objectives, researchers and healthcare professionals can gain a deeper understanding of the links between depression and CVD, leading to improved patient care, early intervention, and better overall health outcomes. In our review, we decided to highlight common risk factors between depression and cardiovascular disease. Particular attention was paid to obesity, diabetes, physical activity, and inflammation. Evidence of a link between depression and cardiovascular disease is still being sought. Studies have been conducted to prove the bi-directional relationship between these diseases. A number of studies have established evidence of cardiovascular autonomic dysfunction in depressed patients. Increased mortality in heart disease, especially in specific sections of the population such as myocardial infarction patients, is also associated with multiple markers of autonomic dysregulation of the cardiovascular system. These include an increase in the resting and circadian heart rate, acceleration or deceleration of the heart rate in response to physical stressors, variability in the pace and sensitivity of baroreceptors, and high variability in ventricular repolarization [24].

In 2008, the Scientific Advisory Committee on Depression and Coronary Heart Disease of Americans, in conjunction with The Heart Association (AHA), recommended screening for depression in patients diagnosed with heart disease [25]. In March 2014, the AHA recommended elevating depression to a risk factor for adverse outcomes in patients with acute coronary syndrome [26].

Matthews et al. (2005) found in their research that there is a relationship between the severity of depressive symptoms and systemic vascular resistance (SVR). In particular, in the SVR, the effect of interaction between the severity of the depressive episode and stress was observed, and the severe course of depression was associated with a significantly higher SVR at rest [27]. Additionally, Whang et al. conducted a large prospective study on the health of nurses in the USA, the results of which showed that the symptoms of depression are associated with a higher probability of subsequent cardiovascular events, and even worse, sudden cardiac death [28]. Interestingly, depressed patients treated for a long time with selective serotonin reuptake inhibitors (SSRIs) had a lower risk of myocardial infarction than depressed patients not taking antidepressants [29].

## 3. Obesity

Overweight and obesity are defined as abnormal or excessive fat accumulation that presents a risk to health. A body mass index (BMI) over 25 is considered overweight, and over 30 is obese. The issue has grown to epidemic proportions, with over 4 million people dying each year as a result of being overweight or obese in 2017 according to the global burden of disease [30]. Obesity is one of the main causes of lifestyle diseases. Depression and obesity often coexist in individuals. They are common diseases with serious public health consequences. The relationship between these conditions is two-way: the presence of one increases the risk of developing the other [31]. Numerous studies have shown a strong association between obesity and the development of depressive symptoms and clinical depression. Several longitudinal studies have also shown a higher incidence of depression among individuals with obesity [32,33]. The relationship between obesity and depression is complex and multifaceted, influenced by various biological, psychological, and social factors [34]. Psychosocial factors play a role in the association between obesity and depression. Stigmatization, discrimination, and low self-esteem related to obesity can contribute to the development of depressive symptoms [35]. It is important to note that not all individuals who are obese will develop depression, and not all individuals with depression will be obese. Shared biological mechanisms may contribute to the relationship between obesity and depression. Chronic low-grade inflammation, dysregulation of the hypothalamic–pituitary–adrenal axis, and altered brain neurotransmitter systems have been implicated in both conditions [36,37]. However, the strong association between the two conditions highlights the need for comprehensive care that addresses both physical and mental well-being when managing obesity and depression. Integrated treatment approaches, including psychological support, lifestyle modifications, and appropriate medical interventions, can help improve outcomes for individuals dealing with obesity and depression. Depression has also been identified as a risk factor for the development of obesity. Depressive symptoms can lead to unhealthy behaviors, such as a sedentary lifestyle, poor dietary choices, and emotional eating, which contribute to weight gain [38]. Obesity can complicate the management of depression. Studies have shown that obesity is associated with a poorer response to antidepressant treatment and a higher risk of treatment resistance [39]. Additionally, obesity-related health problems and physical discomfort may contribute to reduced mobility and impaired self-esteem, affecting the overall well-being of individuals with depression. Depression can hinder weight-management efforts in individuals with obesity. Depressive symptoms can lead to reduced motivation, low energy levels, and decreased adherence to healthy lifestyle behaviors, such as physical activity and dietary modifications [40]. Addressing the underlying depression is essential to support successful weight management efforts.

Obesity is not only a risk factor for depression but also for cardiovascular disease (CVD). Obesity is widely recognized as a significant risk factor for cardiovascular disease (CVD). Excessive body weight, particularly when it is characterized by excess fat accumulation, places strain on the cardiovascular system and increases the likelihood of developing various cardiovascular conditions [31,41]. The relationship between obesity, depression, and cardiovascular disease is complex, with each condition influencing the development and progression of the others. Obesity significantly increases the risk of developing cardiovascular diseases. A meta-analysis performed by Whitlock et al. involving over 900,000 participants showed that obesity was associated with a higher risk of coronary heart disease, stroke, and heart failure [42]. Several other studies have consistently demonstrated the increased risk of hypertension, dyslipidemia, coronary artery disease, and other cardiovascular conditions in individuals with obesity [43,44].

Du et al. in a study showed that the prevalence of depression was 17.83% in patients with central obesity and 12.6% in non-obese patients. Thus, there is a noticeable difference in the incidence of depression between the two groups. This leads to the conclusion that the occurrence of depression is positively related to the degree of obesity [45]. Fox et al. found that depression and anxiety were associated with more severe obesity among adolescents seeking treatment [46].

Obesity contributes to cardiovascular diseases through various mechanisms. Excess adipose tissue leads to the release of pro-inflammatory cytokines and adipokines, which promote systemic inflammation and endothelial dysfunction [47]. Obesity is also associated with insulin resistance, dyslipidemia, and metabolic abnormalities, all of which contribute to the development of atherosclerosis and cardiovascular risk [48,49]. On the one hand, obesity itself contributes to this risk, and on the other hand, the risk is also affected by diseases associated with it, such as hypertension, diabetes, insulin resistance, and sleep apnea [50]. Obesity adversely affects the cardiovascular (CV) system in several ways. It may also depend on the distribution of body fat. These complex obesity issues remain the greatest challenge for clinicians dealing with multiple obesity phenotypes. Due to the prevalence of obesity, physicians and nutritionists should have the skills and tools to recognize high-risk forms of obesity. It is important to quickly identify patients with visceral obesity and patients with severe obesity. Imaging and cardiometabolic studies have clearly shown that reducing BMI lowers also the risk of CVD, at least in overweight or moderately obese patients [51]. Interestingly, a study conducted by Rogge et al. in 2013, in patients with initially asymptomatic aortic stenosis, being overweight or obese had no effect on disease progression or related cardiovascular or ischemic events, but both were associated with increased mortality [52]. In 2019, Pagidipati et al. in their work concluded that those who were overweight or obese class I had a lower cardiovascular risk than those who were underweight/normal weight. These results suggest the presence of an obesity paradox, but this paradox may reflect an epidemiological artifact rather than a true negative relationship between normal body weight and clinical outcomes [53]. Daumit et al. showed that behavioral counseling, care coordination, and care management intervention statistically significantly reduced the overall risk of cardiovascular disease in adults with major mental health conditions [54]. In a study conducted by Faulconbridge et al. the authors tested whether a combination treatment targeting obesity and depression at the same time would produce greater improvements in weight, mood, and CVD risk factors than a treatment targeting each disease individually. The results showed that behavioral weight management resulted in short-term improvements in weight, mood, and CVD risk, comparable to a combination treatment of cognitive behavioral therapy for depression [55]. Weight loss has been shown to have significant cardiovascular benefits. Studies have demonstrated that intentional weight loss through lifestyle modifications or bariatric surgery can improve multiple cardiovascular risk factors, including blood pressure, lipid profiles, and insulin sensitivity [56,57]. Weight-loss interventions have also been associated with a reduction in the incidence of cardiovascular events [58]. The increasing prevalence of obesity worldwide has significant public health implications for cardiovascular disease burden. Addressing obesity through prevention efforts, lifestyle modifications, and interventions targeting weight loss is crucial in reducing the incidence and impact of cardiovascular diseases [59].

## 4. Physical Activity

Physical inactivity is a known risk factor for many diseases, including cardiovascular disease. Regular physical activity is proven to help prevent and manage noncommunicable diseases (NCDs) such as heart disease, stroke, diabetes, and several cancers. It also helps prevent hypertension, maintain healthy body weight and can improve mental health, quality of life, and well-being. Physical activity refers to all movement. Popular ways to be active include walking, cycling, wheeling, sports, active recreation, and play, and can be performed at any level of skill and for enjoyment by everybody [60]. As a preventive measure, it is recommended to exercise three times a week for at least 30 min continuously with a heart rate of at least 130 beats per minute. Thanks to physical activity, it is also possible to relieve stress. Exercise is a physiological stressor that can benefit the cardiovascular system in many ways. The evidence collected so far is sufficient to consider exercise an essential tool in the prevention of cardiovascular disease, if properly prescribed and supervised [61]. Physical inactivity refers to a lack of regular physical activity or exercise. It involves a sedentary lifestyle and a minimal amount of movement or engagement in physical activities. Research has shown that physical inactivity can be considered a risk factor for depression [62]. Sedentary behavior has been associated with an increased risk of developing depression. Individuals who engage in high levels of sedentary behavior, such as excessive TV watching or computer use, have shown higher rates of depression compared to those with lower sedentary behavior levels [63]. It is important to note that while physical inactivity can be considered a risk factor for depression, it is not the sole determinant of the condition. Depression is a complex disorder influenced by a variety of genetic, environmental, and psychological factors. However, incorporating regular physical activity into one’s routine can be beneficial for both physical and mental well-being, potentially reducing the risk of depression and promoting overall health. Sedentary behavior has been associated with negative effects on mental health markers, such as increased levels of stress, anxiety, and symptoms of depression. Prolonged sitting and physical inactivity can disrupt neurotransmitter regulation and impair brain functioning, leading to mood disturbances [64]. Lack of physical activity is a well-established risk factor for cardiovascular diseases (CVD). It is important to note that physical inactivity is just one of several risk factors for CVD, and its impact on cardiovascular health can be influenced by other factors such as genetics, diet, and smoking. However, adopting a physically active lifestyle can significantly reduce the risk of developing CVD and improve overall cardiovascular health [65].

Exercise is considered a non-pharmacological intervention that may delay obesity-related comorbidities, improve cardiovascular fitness, and modulate inflammation. These lead to an improvement in the immune response and the attenuation of low-grade chronic inflammation, characterized by the release of cytokines, which provides benefits at a systemic level. Many studies report that this happens by reducing visceral adipose tissue mass, with a subsequent reduction in the release of adipokines from adipose tissue (AT) and/or by inducing an anti-inflammatory environment [66]. Cattadori et al. provided a comprehensive review of the impact of physical fitness and physical activity on the risk, management and prognosis of heart failure (HF). Exercise is a basic preventive tool in patients with HF. So, exercise training is a form of therapy. Good physical condition, i.e., normal exercise capacity, is a strong prognostic parameter in patients with HF [67].

Lapmanee et al. found in rats that voluntary jogging was effective in reducing anxiety and depression-like behavior. Intense effort and forced effort did not give such a result. On the contrary—they caused stress by intensifying the symptoms of anxiety and depression [68]. Danielsson et al. in their randomized study showed that physical exercise in a physiotherapeutic setting seems to have an effect on depressive severity and performance in major depression. Their findings suggest that physical therapy may be a viable clinical strategy that inspires and guides people with major depression to exercise [69].

Schuch et al. in a meta-analysis showed a strong antidepressant effect of exercise. In studies of participants diagnosed with major depressive disorder (MDD), the positive effect of exercise was greater. This effect was more pronounced in outpatients who did not have other comorbidities and exercised under the supervision of qualified trainers [70]. Soucy et al. (2017) in their work noted that activities such as behavioral activation (BA) and physical activity (PA) can reduce the severity of depressive symptoms in adults. Improvement may persist for up to two months of follow-up. Both types of activity had a large and statistically significant effect. Physical activity was more effective in relieving symptoms. Their findings suggest that various forms of activation, whether physical or daily activity, reduce symptoms of depression [71]. Sedentary behavior has been consistently linked to an increased risk of developing CVD, including coronary heart disease, stroke, and cardiovascular mortality. Sedentary individuals have shown a higher incidence of these conditions compared to those who engage in regular physical activity [72]. Prolonged sitting and sedentary behavior have been associated with detrimental effects on cardiovascular health markers, such as increased blood pressure, unfavorable lipid profiles (elevated triglycerides and reduced high-density lipoprotein cholesterol), impaired glucose metabolism, and increased levels of inflammatory markers [73]. The work of Reed et al. found that for patients with coronary heart disease, exercise programs were well attended, safe, and beneficial in improving physical and mental health [74]. Johansson et al. in their project, they evaluated the effects of a 9-week online cognitive behavioral therapy program compared to an online discussion forum on depressive symptoms in patients with cardiovascular disease. The results showed that in the online cognitive behavioral therapy group, a significant correlation was found between changes in depressive symptoms and changes in physical activity [75]. A study conducted by Peterson et al. showed that patients with severe depressive symptoms who achieved the primary outcome of the study, which was an increase in physical activity of ≥336 kcal/week, had significantly lower rates of cardiovascular morbidity and mortality at 12 months [76].

## 5. Diabetes

Diabetes is a chronic, metabolic disease characterized by elevated levels of blood glucose (or blood sugar), which leads over time to serious damage to the heart, blood vessels, eyes, kidneys, and nerves. The most common is type 2 diabetes, usually in adults, which occurs when the body becomes resistant to insulin or does not make enough insulin. In the past three decades, the prevalence of type 2 diabetes has risen dramatically in countries of all income levels [77]. Diabetes mellitus diabetes management and self-care behaviors, leading to poorer glycemic control and an increased risk of complications. Therefore, addressing both the physical and mental health needs of individuals with diabetes is essential to promote overall well-being and improve outcomes [78]. It is well-established that diabetes is a significant risk factor for cardiovascular diseases (CVD). In fact, individuals with diabetes are at a much higher risk of developing heart disease compared to those without diabetes. Managing diabetes effectively can help reduce the risk of cardiovascular diseases. Controlling blood sugar levels through medication, adopting a healthy diet, engaging in regular physical activity, managing blood pressure and cholesterol levels, and avoiding smoking are key steps in preventing or minimizing the impact of diabetes on cardiovascular health. Regular check-ups with healthcare providers are crucial for monitoring and addressing any potential risks or complications [79]. In developed countries such as the United States (US) and the United Kingdom, many epidemiological studies have been conducted on depression and diabetes and their comorbidities. Existing reports indicate that the situation is also similar in other countries, although it is not as well documented [80]. The prevalence of depressive disorders in diabetics generally ranges from 10% to 15%. This means that there is about twice the incidence of depression in people without diabetes. The coexisting disease significantly worsens the prognosis for both diseases and increases their mortality [81]. Individuals with diabetes are at a higher risk of developing depression compared to the general population. A systematic review and meta-analysis performed by Ali et al. found that the prevalence of depression in people with diabetes was nearly double that of those without diabetes [82]. Other studies have reported similar findings, highlighting the increased vulnerability to depression among individuals with diabetes [83]. Several biological and psychosocial mechanisms contribute to the association between diabetes and depression. The chronic inflammation and oxidative stress associated with diabetes can impact brain function and increase the risk of depressive symptoms [83]. Additionally, the psychosocial stressors related to diabetes management, such as dietary restrictions, medication adherence, and fear of complications, can lead to emotional distress and depression [84]. Depression can have a detrimental effect on the management and outcomes of diabetes. A meta-analysis performed by Gonzalez et al. demonstrated that depression was associated with poorer glycemic control among individuals with diabetes [85]. Depression can hinder self-care behaviors, such as medication adherence, regular physical activity, and healthy eating, leading to suboptimal diabetes control and an increased risk of complications [86]. The presence of diabetes can also complicate the treatment of depression. People with diabetes and comorbid depression may have a poorer response to antidepressant medications compared to those without diabetes [87]. Additionally, diabetes-related symptoms, such as fatigue and decreased motivation, can overlap with depressive symptoms, making it challenging to differentiate and manage both conditions effectively [88]. Diabetes and depression share several common risk factors, including obesity, sedentary lifestyle, and genetic predisposition [89,90]. Joseph and Golden (2016) hypothesized that the dysregulation of the hypothalamic-pituitary-adrenal (HPA) axis is an important biological link between stress, depression, and diabetes. A flatter or blunted circadian cortisol curve that is relatively maintained throughout life is associated with a risk of developing depression. Suppression of the circadian cortisol curve is a specific predictor of diabetes and higher glycated hemoglobin levels in diabetic patients. This variable is an important characteristic of cardiometabolic risk. Dysregulation of the HPA axis has been found to be a critical link in the high incidence of depression and comorbid diabetes [91].

Diabetes significantly increases the risk of developing cardiovascular disease. A meta-analysis performed by Sarwar et al. involving more than 450,000 individuals demonstrated that individuals with diabetes have approximately twice the risk of developing CVD compared to those without diabetes [92]. Several other studies have confirmed this association and have shown that diabetes is an independent risk factor for the development of CVD [93]. Chronic hyperglycemia and insulin resistance, key features of diabetes, contribute to the development and progression of cardiovascular disease. Prolonged exposure to elevated blood glucose levels can damage blood vessels, leading to atherosclerosis and increased risk of coronary artery disease, myocardial infarction, and stroke [94,95]. Insulin resistance, commonly observed in type 2 diabetes, is also associated with endothelial dysfunction and impaired cardiac function [96]. Diabetes and cardiovascular disease share common risk factors, such as obesity, hypertension, dyslipidemia, and a sedentary lifestyle. These risk factors often cluster together and contribute to the development of both conditions [97]. The presence of diabetes can further amplify the impact of these risk factors, leading to a higher cardiovascular risk. Diabetes can also lead to a specific form of heart disease known as diabetic cardiomyopathy. It is characterized by structural and functional changes in the heart muscle, independent of coronary artery disease or hypertension. Diabetic cardiomyopathy is associated with impaired cardiac function, diastolic dysfunction, and an increased risk of heart failure [98]. Medications commonly used for CVD management, such as beta-blockers and thiazide diuretics, can affect glycemic control and insulin sensitivity, requiring adjustments in diabetes treatment [99]. Furthermore, the presence of CVD can complicate self-care behaviors, such as physical activity, leading to difficulties in glycemic control and diabetes management [100]. Given the strong association between diabetes and cardiovascular disease, comprehensive management strategies should address both conditions simultaneously. Lifestyle modifications, including healthy eating, regular physical activity, and weight management, are crucial for reducing cardiovascular risk in individuals with diabetes [101]. Additionally, aggressive management of cardiovascular risk factors, such as blood pressure, cholesterol, and glycemic control, is essential to prevent or delay the onset of CVD complications [102].

## 6. Inflammation

Inflammation is the body’s natural response to protect itself from damage. It is the body’s immune system response to an irritant. There are two types: acute and chronic [103]. Inflammation is a complex physiological response by the immune system to protect the body from harmful stimuli, such as pathogens, toxins, or tissue damage. While inflammation is a crucial defense mechanism, chronic or persistent inflammation can have negative effects on various aspects of health, including mental health. There is increasing evidence suggesting that inflammation may play a role in the development and progression of depression. The presence of chronic inflammation can increase the risk of developing depression, particularly in individuals who may be genetically predisposed or have other risk factors. Understanding the relationship between inflammation and depression can potentially lead to new treatment approaches targeting inflammation as a way to manage or prevent depressive symptoms [104]. Inflammation is increasingly recognized as a significant risk factor for cardiovascular disease (CVD). While inflammation is a natural response of the immune system to injury or infection, chronic inflammation can contribute to the development and progression of various cardiovascular conditions. Inflammation has also been implicated in the development and progression of other cardiovascular conditions such as peripheral artery disease (narrowing of blood vessels outside the heart), heart valve disease, and arrhythmias [105]. Managing inflammation is crucial for reducing the risk of cardiovascular disease. In some cases, anti-inflammatory medications may be prescribed by healthcare professionals to manage chronic inflammation associated with specific conditions. Early detection and intervention are essential to prevent or mitigate the inflammatory processes that contribute to cardiovascular disease. However, when inflammation becomes chronic or lasts too long, it can prove harmful and can lead to disease. The role of pro-inflammatory cytokines, chemokines, adhesion molecules, and inflammatory enzymes has been linked to chronic inflammation. Chronic inflammation has been found to mediate a wide variety of diseases, including cardiovascular disease, cancer, diabetes, arthritis, Alzheimer’s disease, lung disease, and autoimmune disease [106]. Inflammation is a biological process that protects the body from threatening factors in order to maintain homeostasis. Excessive inflammation contributes to the pathophysiology of various diseases [107].

Many modern studies show that even low levels of chronic inflammation contribute to depression. From a scientific point of view, depression is increasingly often defined as a complex pathophysiological condition associated with excessive inflammation. Elevated levels of pro-inflammatory cytokines in patients have been cited as evidence. Another important aspect is the cellular response of the immune system. In the body of people with this mood disorder, mechanisms related to inflammation occur, which results in an increase in the activity of platelet activating factors, oxidative and nitrogen stress, and mitochondrial dysfunction. This information may initiate new trends in treatment [108]. The inflammatory process and the secondary activation of the immune system in depression are seen in the peripheral and central nervous systems. This explains the relationship between immune and inflammatory mood disorders [109]. Moludi et al. in studies tested the anti-inflammatory and antidepressant effect of Lactobacillus Rhamnosus G (LGG), a probiotic strain, alone or in combination with the prebiotic, inulin, in patients with ischemic heart disease. The effects of synbiotics have been shown to control both chronic inflammation and depression. These results suggest that a probiotic plus a prebiotic may exert at least some of their effect on depression through inflammatory cytokines [110]. Kiecolt-Glaser et al. decided to investigate increased intestinal permeability (“leaky gut”) as one potential mechanistic pathway from marital distress and depression to increased inflammation. Two endotoxin biomarkers, LPS-binding protein (LBP) and soluble CD14 (sCD14), as well as C-reactive protein (CRP), interleukin 6 (IL-6) and tumor necrosis factor alpha (TNF-α) were used to assess inflammation. Particits with more hostile marital interactions were shown to have higher LBP than those who were less hostile. These results indicate that, among other things, a difficult marriage and a history of mood disorders may promote a pro-inflammatory environment through increased intestinal permeability, thus fueling inflammation-related disorders [111].

Research conducted in 2007 by Vaccarino et al. showed a strong relationship between depression and biomarkers of inflammation, and partly explains the link between depression and cardiovascular disease [112].

Cardiovascular health deteriorates with age, and age is one of the strongest risk factors for cardiovascular complications, including myocardial infarction, heart failure, arrhythmias, and heart-related death. An important risk factor for complications of cardiovascular diseases is age. The expression of pro-inflammatory cytokines increases throughout human life. Their increased concentrations are not only markers of chronic low-grade inflammation, but also affect the cardiovascular system. They promote autonomic and sympathetic nervous-system imbalance, increase myocardial electrical instability, stimulate remodeling and inhibit cardiac function, accelerate endothelial dysfunction, vasoconstriction, and atherosclerosis progression. Additionally, they impair kidney function. Through these mechanisms, the cardiovascular system ages faster and, consequently, increases its susceptibility to cardiovascular morbidity and death [113]. The findings of Zhou et al. suggest that systemic inflammation in patients with heart failure is causally related to the function of mitochondria in peripheral blood mononuclear cells [114]. Ridker et al. conducted a joint analysis of patients with or at high risk of atherosclerotic disease who were receiving modern statins and were participants in international trials. It was shown that among patients receiving modern statins, inflammation assessed by high-sensitivity CRP was a stronger predictor of the risk of future cardiovascular events and death than cholesterol assessed via LDLC. These data influence the choice of adjuvant therapy in addition to statin therapy and suggest that the combination of aggressive lipid-lowering and anti-inflammatory therapies may be necessary to further reduce the risk of atherosclerosis [115]. Koenig et al. used a sensitive immunoradiometric assay to investigate the association of serum C-reactive protein (CRP) with the rate of first major ischemic heart event (CHD). There was a positive and statistically significant unadjusted relationship between CRP values and the incidence of CHD events. These results confirm the prognostic importance of CRP for CVD risk. They suggest that low-grade inflammation, and especially its thrombo-occlusive complications, are involved in the pathogenesis of atherosclerosis [116]. Gusev et al. reviewed research on the underlying cause of atherosclerosis, noting that atherosclerosis was not initially thought of as an inflammatory disease. Many authors subsequently believed it to be a chronic, low-grade inflammatory condition. There is now ample evidence that the formation of atherosclerosis more closely resembles classical variants of productive inflammation involving various immune-response vectors [117]. Garcia-Arellano et al. used the Dietary Inflammatory Index (DII) in their research to assess the inflammatory potential of nutrients and foods in the context of dietary pattern. In their study, they prospectively investigated the relationship between DII and CVD. The results provide direct prospective evidence that a pro-inflammatory diet is associated with a higher risk of clinical cardiovascular events [118]. Sandoo et al. showed in their studies that classic CVD risk may affect endothelial function more than disease-related inflammatory markers in rheumatoid arthritis. Classic CVD risk factors and anti-TNF-α drugs have different effects on microvascular and macrovascular endothelial function, suggesting that combined CVD prevention strategies may be necessary [119].

Microglia are specialized immune cells that make up 5–10% of all brain cells and perform functions similar to macrophages and other specialized cells [120]. It consists of macrophage cells that reside in the central nervous system. They have the ability to migrate to all areas of the central nervous system through the brain parenchyma. They develop a specific, branched morphological phenotype known as “resting microglia”. A large number of signaling pathways allow it to communicate with macroglia as well as neurons and cells of the immune system. Microglial cells are the most sensitive sensors of brain pathology. They activate when signs of brain damage or nervous system dysfunction are detected. The “activated microglial cell” has the ability to release a large number of substances that can be harmful or beneficial to surrounding cells. They are also able to migrate to the site of damage, proliferation and phagocytosis of cells and cell compartments [121]. These processes ultimately lead to the production of pro-inflammatory cytokines by microglial cells. This event requires two events with different timing: the activation of a fast afferent neural pathway and the slower propagation of the cytokine message in the brain. This is likely to sensitize target brain structures to the production and action of cytokines that spread from the periventricular organs and choroid plexus to the brain. The peripheral innate immune response of the brain is similar in many respects to the peripheral response. The difference is that this brain response does not involve the invasion of immune cells into the parenchyma and is not distorted by tissue damage at the site of infection [122]. Macrophages play a key functional role in the pathogenesis of various cardiovascular diseases, such as atherosclerosis and aortic aneurysms. Their accumulation in the vascular wall leads to a persistent local inflammatory response characterized by the secretion of chemokines, cytokines, and enzymes that degrade matrix proteins [123].

Minocycline, which inhibits microglial activation, represents a promising diversion candidate for the treatment of treatment-resistant depression. Therefore, this theory was tested by Hellmann-Regen et al. Interestingly, minocycline 200 mg/d was added to antidepressant treatment for 6 weeks. Minocycline was well tolerated but no better than a placebo in reducing depressive symptoms. The results of this work highlight the unmet need for therapeutic approaches and predictive biomarkers in drug-resistant depressive disorders [124]. Additionally, a study conducted by Hasebe et al. aimed to investigate the effect of minocycline adjuvant treatment on inflammatory and neurogenesis markers in major depressive disorder (MDD). Serum samples were collected from a randomized, placebo-controlled 12-week clinical trial of minocycline (200 mg/day, added as usual to treatment) in adults experiencing MDD to determine changes in interleukin-6 (IL-6), lipopolysaccharide binding protein (LBP) and brain-derived neurotrophic factor (BDNF). There was no difference between the adjuvant minocycline or placebo groups at baseline or week 12 in IL-6, LBP, or BDNF levels. Interestingly, higher levels of IL-6 at the start of the study were predictive of greater clinical improvement. Exploratory analyzes suggested that a change in IL-6 levels was significantly associated with anxiety symptoms and quality of life [125].

The blood–brain barrier (BBB) is a kind of link between the plasma and the brain. Its task is to prevent the entry of neurotoxic plasma components, blood cells and pathogens into the brain and to regulate the transport of molecules to and from the central nervous system (CNS), which maintains a strictly controlled chemical composition of the entire nervous system and an environment that is necessary for the proper functioning of neurons [126,127]. BBB transport is selective. Some of the transported substances are also some cytokines. These include interleukin (IL)-1α and IL-1β, IL-1 receptor antagonist (IL-1ra), IL-6, tumor necrosis factor-α (TNF), leukemia inhibitory factor (LIF) and ciliated neurotrophic factor, and many adipokines. They play an important role in the physiological response to inflammation and neuro-regeneration. Cytokines are known to be associated with autoimmune diseases, infections, trauma-related inflammation, ischemia, hemorrhage, neurodegeneration, and some genetic disorders. Not without significance is their impact on brain physiology, including eating behavior (Table 3), sleep, thermoregulation, emotions, and memory [128]. Cytokines such as IL-1, IL-6, IL-10, and TNF-alpha also play a major role in the inflammatory processes underlying cardiovascular disease [129].

Kruse et al. attempted to test whether IL-8 predicts a depressive response to ketamine and whether it depends on participants’ gender. Plasma IL-8 levels were assessed at baseline and post-treatment. The change in IL-8 levels from baseline to post-treatment differed significantly by response status (defined as a ≥50% decrease in the Hamilton Depression Rating Scale) by gender. Increasing IL-8 was associated with a decrease in HAM-D score in women, while the opposite was found in men [89]. In another study in 2022 by another team, Kruse et al. investigated whether higher levels of IL-8 attenuated increases in depressed mood in response to an experimental model of inflammation-induced depression. Given the epidemiological associations identified between IL-6, tumor necrosis factor (TNF)-α, and subsequent depression, the levels of these pro-inflammatory cytokines have also been studied as potential moderators of the depressed mood response to endotoxins. Their findings suggest that IL-8 may be a biological agent that reduces the risk of inflammation-related depression [133]. The aim of Yang et al. was to check whether the level of pro-inflammatory cytokines in the serum is correlated with the development of post-stroke depression. The concentration of pro-inflammatory cytokines (IL-6, IL-18 and TNF-alpha) in the serum of all patients was determined on the 1st and 7th days after admission. Serum IL-18 concentration on day 7 was significantly higher in patients with post-stroke depression than in patients without post-stroke depression. This may suggest that serum IL-18 determination on day 7 after admission may predict the risk of post-stroke depression both in the acute stage of stroke and 6 months after stroke [134]. Depressed people are prone to sleep disturbances, which in turn can perpetuate depression. The aim of the study conducted by Siu-Man et al. was to evaluate the effect of these two mind–body therapies on people with depressive symptoms and sleep disorders. The outcome measures were plasma IL-6 and IL-1β concentrations and a questionnaire including the Pittsburgh Sleep Quality Index, the Center for Epidemiology Research Depression Scale, the Somatic Symptoms Inventory, the Perceived Stress Scale, and the Holistic Body–Mind–Spirit Well-Being Scale. The study showed a bidirectional relationship between depression and sleep disorders, and a significant effect of depression and sleep disorders on IL-6 and IL-1β [135].

Increased inflammation alerts organisms to danger. The immune system produces cytokines by affecting the CNS, a complex mediated by pro-inflammatory cytokines. (Figure 1). Common immune-inflammatory pathways underlie the physiology of sickness behavior and the clinical pathophysiology of depression. This relationship is two-way [136]. The main cause of cardiovascular disease is atherosclerosis. It is a chronic inflammatory disease of blood vessels. Various inflammatory cells and inflammatory factors are believed to play a significant role in its pathogenesis. The pathological response to various vascular wall lesions induces classical inflammation that leads to degeneration, exudation, and hypertrophy. The detection of inflammatory biomarkers, such as CRP or IL-6 adhesion molecules, may be a good way of diagnosing atherosclerosis and cardiovascular disease [137]. Scientists are still looking for the causes of civilization diseases, such as depression and atherosclerosis. Many cytokines and their levels have been studied in patients suffering from the diseases mentioned above. Elevated levels of the same cytokines in these disease entities may suggest links between depression and atherosclerosis. A brief summary of examples of cytokines associated with both depression and atherosclerosis is provided in Table 4.

In the above paragraphs, the authors have focused on possible factors, including obesity, inflammation, diabetes and physical activity constituting links between depression and CVS. A detailed review of each has been presented along with a possible linking pathway between depression and CVDs.

## 7. Discussion

As we can see, many authors and researchers have been looking for answers to questions about the relationship between depression and CVD for a very long time. Inflammation seems to be the strongest variable influencing these relationships. Further research in this area should be considered. There are a number of factors to consider when interpreting scientific publications and research results. First, depression and cardiovascular disease are very complex issues in terms of medical knowledge and their social consequences. Many studies do not control for potential confounders, and most of the literature is cross-sectional. A growing body of the literature is focusing on the interaction between depression and inflammatory diseases, which include most CVDs. The number of patients visiting a doctor with a complex set of overlapping symptoms, including emotional and physical ailments, including stenocardia, is growing. Physical ailments usually do not indicate the cause of the physical condition of the patient. Factors linking depression and cardiovascular disease has been researched worldwide for many years. In our narrative review, we looked for evidence of comorbidity between depression and CVD. For many years, factors linking depression and cardiovascular disease have been studied all over the world. We found many studies that showed a positive correlation between depression and CVD. The results showed that people with CVD are more likely to suffer from depression than the general population. Common risk factors for these diseases are also evident, such as obesity, physical inactivity, diabetes, and inflammation. The scheduled state seems to be the most widely studied issue here. Reports have indicated a significant association between CVD and depression. Depression is a mental health disorder characterized by persistent feelings of sadness, loss of interest, and other emotional and physical symptoms. CVD refers to a range of conditions affecting the heart and blood vessels, including coronary artery disease, heart failure, and stroke. Studies have consistently shown that individuals with depression have a higher risk of developing CVD. The relationship between the two conditions is bidirectional, meaning that depression can contribute to the development of CVD, and having CVD can increase the risk of developing depression. Depression can lead to unhealthy behaviors such as poor diet, sedentary lifestyle, smoking, and non-adherence to medication, which are risk factors for CVD. Additionally, depression is associated with physiological changes in the body, including inflammation, hormonal imbalances, and increased sympathetic nervous system activity, all of which can contribute to the development and progression of CVD. Conversely, individuals with CVD are more prone to developing depression due to the psychological and emotional impact of dealing with a chronic illness. The physical limitations imposed by CVD, the fear of future cardiac events, and the disruption of daily life can lead to feelings of hopelessness, sadness, and anxiety. Both depression and CVD share common underlying mechanisms, such as dysregulation of the hypothalamic-pituitary-adrenal axis, increased oxidative stress, and endothelial dysfunction. These shared pathways may help explain the strong association between the two conditions. Recognition of the link between depression and CVD is crucial for healthcare providers. Identifying and treating depression in individuals with CVD can improve their quality of life and potentially reduce the risk of further cardiovascular events. Similarly, addressing cardiovascular risk factors in individuals with depression can help prevent the development of CVD. Integrative approaches that combine pharmacological interventions, psychotherapy, lifestyle modifications, and social support have shown promise in managing both depression and CVD. Collaborative care models, where primary care physicians, cardiologists, and mental health professionals work together, have been effective in improving outcomes for patients with comorbid depression and CVD. Depression and cardiovascular disease are closely linked, with each condition increasing the risk of the other. Recognizing and addressing this association is important for providing comprehensive care to individuals affected by these conditions. Further research is needed to better understand the mechanisms underlying the relationship and to develop more targeted interventions. Research suggests a strong link between depression and obesity. People with depression are more likely to be obese, and individuals who are obese have a higher risk of developing depression. The exact nature of this association is still under investigation. Factors such as emotional eating, sedentary behavior, and hormonal imbalances may contribute to the development of obesity in individuals with depression. Similarly, the psychosocial consequences of obesity, such as body image dissatisfaction and social stigma, may increase the risk of depression. Obesity is a well-established risk factor for cardiovascular disease. Excess body weight, particularly abdominal obesity, is associated with an increased likelihood of developing conditions such as hypertension, dyslipidemia, and type 2 diabetes, all of which contribute to the development of cardiovascular disease. Obesity-induced inflammation, insulin resistance, and adverse metabolic changes play significant roles in the progression of CVD. Lifestyle modifications, including weight loss, are crucial in reducing the risk of cardiovascular events in obese individuals. Overall, the reports highlight the interconnectedness of depression, cardiovascular disease, and obesity. There is substantial evidence indicating a link between depression and inflammation. Depressed individuals often exhibit elevated levels of pro-inflammatory markers, such as C-reactive protein (CRP), interleukin-6 (IL-6), and tumor necrosis factor-alpha (TNF-alpha). The relationship between depression and inflammation is thought to be bidirectional. Chronic inflammation can contribute to the development of depression, while depression-related alterations in the stress-response system and neurotransmitter imbalances can promote inflammation. Inflammatory processes may impair the functioning of neurotransmitters such as serotonin, which plays a crucial role in mood regulation. This disruption can contribute to the development and maintenance of depressive symptoms. Inflammation is recognized as a significant contributor to the development and progression of CVD. Conditions such as atherosclerosis, which underlies many CVDs, involve an inflammatory response within arterial walls. Inflammatory markers, including CRP, IL-6, and TNF-alpha, have been associated with an increased risk of CVD. These markers can predict the likelihood of future cardiovascular events and help assess the effectiveness of treatment. Chronic inflammation can contribute to the formation of arterial plaques, increase blood clot formation, and impair endothelial function, all of which are important factors in CVD pathogenesis. Depression, CVD, and inflammation are interconnected, with each influencing the other. Depression can contribute to the development of CVD through behavioral and physiological mechanisms, while inflammation plays a significant role in both depression and CVD. Recognizing and addressing the links between these conditions is essential for effective prevention and treatment strategies.

Depression and cardiovascular disease are two prevalent health conditions that can significantly impact an individual’s well-being. Research has indicated a complex and bidirectional relationship between these two conditions, suggesting that they can influence each other’s development and progression. Several studies have established a link between depression and an increased risk of developing cardiovascular disease. Similarly, a meta-analysis performed by Nicholson et al. demonstrated a 64% increased risk of cardiac events in depressed individuals [147]. A systematic review and meta-analysis performed by van Melle et al. analyzed 20 studies and found that depression was associated with a two-fold increased risk of mortality in patients with coronary artery disease. Another meta-analysis performed by Thombs et al. involving 22 studies [148]. concluded that depression was associated with a 31% increased risk of adverse cardiovascular events in patients with heart disease [149]. Meng et al. conducted a prospective study of 512,712 adults (302,509 women and 210,203 men) aged 30–79 to assess depression as a risk factor for all-cause and cardiovascular mortality. The results showed that depression was associated with an increased risk of all-cause and cardiovascular mortality in Chinese adults, especially males [150]. On the other hand, cardiovascular disease can also contribute to the development of depression. Patients with cardiovascular disease often experience physical limitations, chronic pain, and reduced quality of life, which can lead to depressive symptoms. A longitudinal study conducted by Lett et al. observed that individuals with heart disease had a two-fold increased risk of developing depression compared to those without heart disease [151]. The underlying mechanisms linking depression and cardiovascular disease are still being investigated. One proposed pathway is the dysregulation of the autonomic nervous system, characterized by increased sympathetic activity and decreased parasympathetic activity, which can contribute to both depression and cardiovascular dysfunction [152]. Depression can negatively affect cardiovascular disease outcomes through various mechanisms. Behavioral factors such as poor medication adherence, unhealthy lifestyle choices, and increased rates of smoking and alcohol consumption have been associated with depression, which can further exacerbate cardiovascular disease [153]. Additionally, depression is associated with physiological alterations, including increased inflammation, platelet activation, and autonomic dysregulation, which can contribute to adverse cardiovascular outcomes [154,155]. Effective management of depression in patients with cardiovascular disease is essential for improving outcomes. Collaborative care interventions, which involve integrating mental health care into the cardiovascular disease management process, have shown promise in reducing depressive symptoms and improving cardiovascular outcomes [156]. Bucciarelii et al. in their work concluded that the presence of depression may worsen cardiovascular morbidity and mortality. They suggest that awareness among cardiologists should be promoted. They pay attention to and emphasize gender issues in order to provide specific answers to male and female patients with CVD due to the different etiology and course of these diseases [157]. Recent studies have begun to address gender and individual differences in susceptibility to both disorders. It is generally believed that the predominance of women in depression is widespread and significant. Parker and Brotchie decided to test these theories in their review. They concluded that while external factors play a role, it is concluded that there is a higher order biological factor (variably defined neuroticism, “stress response” or “limbic overactivity”) that fundamentally contributes to sex differentiation in some manifestations both depression and anxiety, and reflects the impact of steroid changes in the gonads during puberty. Instead of concluding that “anatomy is destiny”, a model of blemish stress, taking into account varied epidemiological outcomes, will emphasize the importance. Ref. [158] Salk et al. found in their meta-analysis that the gender difference in depression is a health disparity, especially during adolescence; however, the magnitude of this difference indicates that depression in men should not be underestimated, as the gender gap peaked during adolescence but then declined and remained stable in adulthood. International analyzes found that greater gender differences were found in countries with greater gender equality for major depression, but not for depressive symptoms [159]. There is much talk about CVD risk factors such as hypertension, dyslipidemia, diabetes, obesity, and smoking. However, despite differences in CVD risk between men and women, most studies evaluating the magnitude of the effect of each risk factor have traditionally focused on men [160]. For many years, female participation in clinical trials was minimal, resulting in a lack of gender-specific analysis of clinical trial data, and, therefore, no specific assessment of risk factors among women. However, scientific advances in the last decade have identified a spectrum of risk factors for cardiovascular disease that may be specific to women. These risk factors, which may include menopause, hypertension, pregnancy disease, and depression, pose additional risks to women beyond traditional risk factors. The current state of knowledge and awareness about these risk factors is currently suboptimal [161]. It is still not known whether the same mechanisms affect sensitivity and immunity in women as in men. Obtaining more information on gender and individual differences in susceptibility to depression and CVD would contribute to both better prevention and treatment [162]. Depression and cardiovascular disease are two prevalent health conditions that can have a significant impact on an individual’s well-being. While they differ in their manifestations, there are several common risk factors that contribute to the development of both conditions. Hare et al. in their clinical review also emphasize the importance of the co-occurrence of systemic disease and depression. It is a common comorbidity that is associated with higher rates of mortality and morbidity. They note that there is sufficient evidence to support the introduction of exercise, talk therapies, and antidepressants to reduce depression in patients with CVD [163]. Meanwhile, lifestyle recommendations in the form of sufficient physical activity and dietary modification can be an invaluable, safe, and useful tool in the treatment of depression, cardiovascular disease and many related immunometabolic disorders [164]. Engaging in regular exercise has been shown to have positive effects on mental health and cardiovascular health. Exercise is efficacious in treating depression and depressive symptoms and should be offered as an evidence-based treatment option focusing on supervised and group exercise with moderate intensity and aerobic exercise regimes [165,166]. Unhealthy eating patterns, such as consuming a diet high in processed foods, added sugars, and saturated fats, are linked to an increased risk of depression and cardiovascular disease [167,168]. Conversely, a diet rich in fruits, vegetables, whole grains, and lean proteins has been associated with a lower risk of both conditions. Obesity is a significant risk factor for both depression and cardiovascular disease. Studies have demonstrated a bidirectional relationship between obesity and depression, with each condition increasing the risk of the other [169,170]. Moreover, obesity is strongly linked to an elevated risk of developing cardiovascular diseases [171]. Chronic stress plays a crucial role in the development of both depression and cardiovascular disease. Prolonged exposure to stress can lead to the dysregulation of various physiological systems, including the immune and cardiovascular systems, and can contribute to the onset and progression of these conditions [172,173]. Joynt et al., in the conclusions of their work, gave theories that stress may be the primary trigger that leads to the development of both depression and cardiovascular disease [174]. However, not every stressed person becomes depressed or suffers from CVD. Dudek et al. emphasize the need to identify not only the biological determinants of susceptibility to stress, but also resilience. Based on the reverse translation approach, rodent depression models were developed to investigate the mechanisms underlying susceptibility and resistance. Both innate and adaptive hormonal and immune responses are enhanced in depressed individuals and in mice exhibiting depressive-like behavior. Neurovascular health is receiving increasing attention as patients with depressive disorders are more likely to have cardiovascular disease, and inflammation is associated with depression, treatment resistance, and relapse [162]. It is important to note that while these risk factors are associated with an increased likelihood of developing depression or cardiovascular disease, they do not guarantee the onset of these conditions. Other factors, such as individual susceptibility and environmental influences, also contribute to the overall risk. Glassman et al. in their work present studies that clearly showed that depression is a risk factor for both incident and recurrence of ischemic heart disease diseases. They also drew attention to the fact that biological factors modifiable health behaviors, especially physical ones, among which inactivity, smoking, and non-compliance seem to be the most critical mediators. This outcome may occur because once a person develops depression, especially if it is recurrent, the illness triggers a series of health behaviors that will undoubtedly increase the risk of vascular disease. It has been repeatedly shown that people with depression take less care of their health, exercise less often, are more likely to be obese and have difficulty quitting smoking [175]. The prospect of integrating the treatment of depression and CVD on the basis of their common pathophysiological elements brings hope. However, there is still not enough research on them. Halaris, in his article, summarizes the evidence showing that the aforementioned co-morbidity is well established, though the relationship between these two serious conditions is complex and multi-faceted. Based on the available literature, inflammation has become a dominant theme and is considered a major mechanism contributing to co-morbidity. An active inflammatory process is present during the active stages of either disease, possibly preceding the onset of debilitating symptoms, and possibly extending beyond mood normalization and symptom remission [176].

Our narrative review reveals how many papers present correlations between depression and CVD. Scientific studies recognize that there is a strong link between the two conditions. Although correlation does not imply causation, the relationship between depression and CVD is complex and involves various factors. Here are some correlations and factors related to depression and CVD (Table 5).

Our review of common risk factors for depression and cardiovascular disease provides valuable insights into the relationship between the two conditions and could potentially aid in the development of prevention strategies and interventions. This study could improve our understanding of common risk factors between depression and cardiovascular disease. Identifying common factors can help scientists and healthcare professionals develop a more comprehensive approach to prevention and treatment. By identifying common risk factors, the study has the potential to facilitate early intervention strategies targeting both depression and cardiovascular disease. This multifaceted approach could lead to better health outcomes and improved quality of life for those at risk. Depression and cardiovascular disease are often treated separately in healthcare systems. Studying common risk factors can promote a more integrated and holistic approach to healthcare by recognizing the link between mental and physical health. It can also inform the development of public health campaigns, policies, and interventions to reduce the incidence and burden of both depression and cardiovascular disease.

However, there are also limitations to this study. For example, establishing causality in observational studies can be difficult. The study may find correlations between some risk factors and depression and cardiovascular disease; however, determining whether these risk factors directly contribute to the development of these conditions requires further research.

Moreover, findings from a particular study may not be universally applicable. Factors such as population demographics, cultural differences, and regional differences in healthcare systems may limit the generalizability of results in other populations. Numerous confounding variables may influence the associations between risk factors, depression, and cardiovascular disease. Factors such as socioeconomic status, genetics, and lifestyle can complicate the interpretation of results. Conducting a mental and physical health examination may also raise ethical concerns. Researchers need to protect participants’ confidentiality, provide adequate support to people with mental health issues, and address any potential stigma associated with depression. Overall, while research into common risk factors for depression and cardiovascular disease has the potential to provide valuable insights, researchers must overcome limitations and challenges to obtain meaningful and reliable results.

Civilization diseases are closely related through common risk factors. It is easy to see that this relationship is bidirectional (Figure 2). Many studies present similar results leading to the conclusion that the relationship between somatic diseases and mood disorders is close.

Being overweight or obese is associated with an increased risk of both depression and cardiovascular disease, and physical activity is a protective factor against the development of these disorders. The morbidity of mood disorders and CVD in diabetics is higher than in the general population (Table 6). Underlying both of these disorders, the inflammatory process heals [93].

It is important for a healthy society to prevent, identify, and treat health problems. The World Health Organization warns us of an increase in both cardiovascular disease and depressive disorders. These measures will probably continue to increase due to, among others, lifestyle and an increase in life expectancy in highly developed countries. The co-occurrence of mental and physical disorders is a major challenge in healthcare around the world. There are two ways to fight civilization diseases, treat or prevent them. This issue seems to be very important and requires further research. Based on our review, there is no doubt that there is a well-established link between cardiovascular disease (CVD) and depression. Studies have consistently shown that people with depression have a higher risk of developing cardiovascular disease, and people with cardiovascular disease have a higher risk of developing depression. Depression is a common mental health condition that can lead to behavioral changes such as reduced physical activity, poor diet, and increased smoking or alcohol consumption. These behavioral changes can increase the risk of developing further disorders, such as obesity and diabetes, and cardiovascular diseases, such as heart attack, stroke, and heart failure. In addition to these behavioral changes, depression has also been linked to biological changes in the body that may contribute to CVD. For example, depression can cause inflammation, which is a key factor in the development of atherosclerosis (a buildup of plaque in the arteries) [158]. Depression can also cause changes in the autonomic nervous system, which can lead to increased heart rate and blood pressure. An interesting direction for further research could be the issue of perceived stress and anxiety related to CVD and their condition, which in turn may lead to the development of depression. Patients may also feel isolated or restricted in their activities, which can contribute to feelings of depression. Treatment is mainly based on pharmacotherapy and interestingly, many authors focus on anti-inflammatory potential of antidepressants [159,160]. Worth clinical attention is also potential treatment targets in CVD [161,162,163]. Prevention should consist of appropriate social education on physical activity and proper diet, preventive programs and screening tests. It is important for healthcare professionals to remember to educate patients about the consequences of leading a harmful lifestyle at each visit [175]. Also, worth attention is the fact that observed associations are present worldwide [144] and probably somehow associated with genetics [176]. Interestingly, depression, CVDs, and systemic blood hypertension are also linked, which seems to be a promising line of future clinical consideration [176] and highlights even more strongly the importance of early detection and diagnosis [162].

It is considered appropriate to conduct further research in this area. It would be worthwhile for further research to focus, among other things, on screening for depression in people with CVD and vice versa. Conducting a study on the relationship between depression and cardiovascular disease can provide valuable insights into the potential link between these two conditions. However, it is important to consider the pros and limitations associated with such studies (Table 7). It is important to acknowledge these pros and limitations when designing and interpreting studies on depression and cardiovascular disease. By addressing methodological challenges and ethical considerations, researchers can contribute to a better understanding of the relationship between these conditions and ultimately improve patient care and public health outcomes.

## 8. Conclusions

There are two ways to fight civilization diseases, treat or prevent them. Clinicians should be prepared to manage comorbid depression and cardiovascular disease. It is very important to look at the patient holistically. This paper reviews the epidemiology and major risk factors for the coexistence of depression and cardiovascular disease. Specific areas have been highlighted that need to be addressed in order to reduce problems arising from the co-existence of the two conditions. This issue seems to be very important and requires further research. Overall, the relationship between CVD and depression is complex and bidirectional. The effective treatment of depression and other psychiatric disorders can be an important element of the prevention and treatment of CVD. Conversely, the effective treatment of CVD can also help prevent depression and improve mental health outcomes. Particular attention should be paid to this during the work of a practicing physician. It seems appropriate to introduce screening tests for depression in patients treated for cardiac reasons. Importantly, in patients treated for mood disorders, it is worth controlling CVD risk factors. Prophylactic blood pressure measurements and heart rate control should be routine at each medical visit.

## Figures and Tables

**Figure 1 biomedicines-11-01903-f001:**
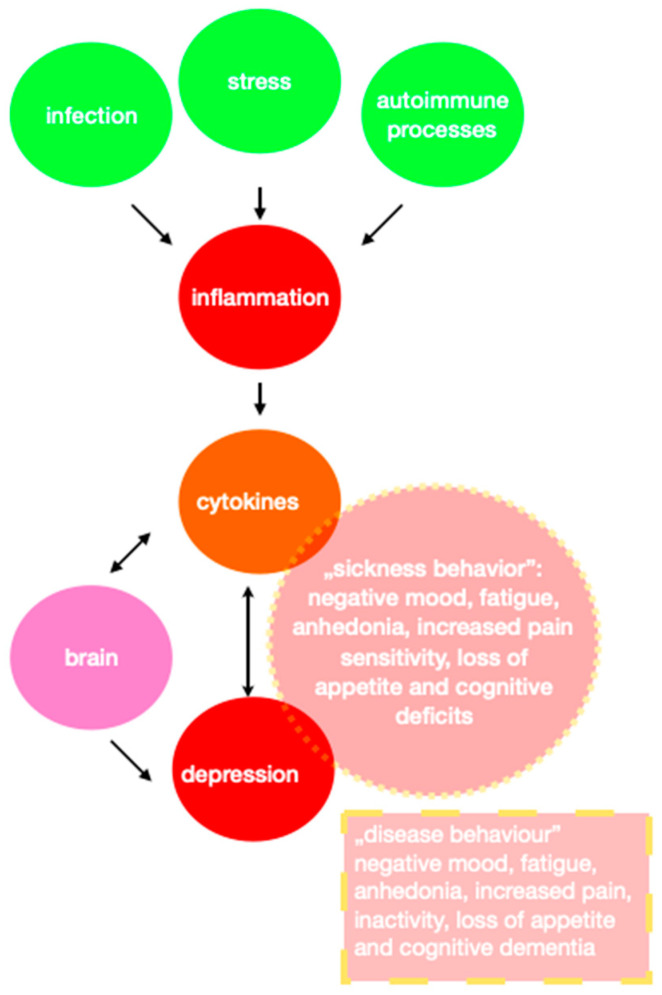
The common mechanism of inflammation and depression.

**Figure 2 biomedicines-11-01903-f002:**
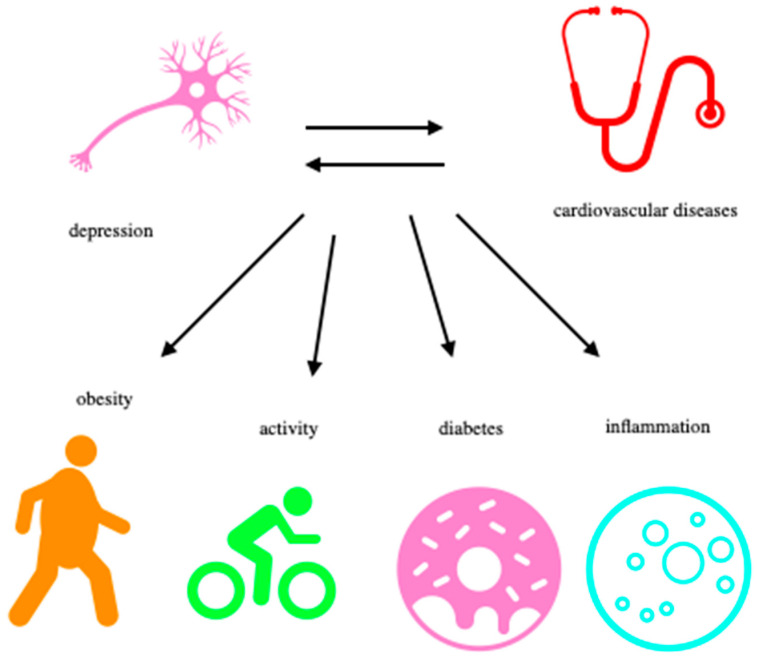
Two-way relationship between depression and cardiovascular disease.

**Table 1 biomedicines-11-01903-t001:** Common symptoms of depression.

Depressed mood, or loss of interest or pleasure.
Depressed most of the day, almost every day, as indicated by subjective feelings (e.g., feeling sad, empty, hopeless) or observations of others (e.g., crying).
Markedly decreased interest or pleasure in all or nearly all activities most of the day, almost every day (according to subjective report or observation).
Significant weight loss when not dieting or gaining weight (e.g., change of more than 5% of body weight in a month) or a decrease or increase in appetite almost every day.
Insomnia or hypersomnia almost daily. Psychomotor agitation or retardation almost daily (observed by others, not just a subjective feeling of restlessness or slowness).
Fatigue or loss of energy almost every day.
Feelings of worthlessness or excessive or inappropriate guilt (which may be delusional) almost daily (not just remorse or guilt about illness).
Decreased ability to think or concentrate, or indecisiveness almost every day (according to subjective accounts or observed by others).
Recurrent thoughts of death (not just fear of death), recurrent suicidal thoughts without a specific plan, or suicide attempts or a specific plan to commit suicide. Symptoms cause clinically significant distress or impairment in social, occupational, or other important areas of functioning.

**Table 2 biomedicines-11-01903-t002:** Objectives of the review article.

The questions:
Do depression and CVD have common risk factors?
Are depression and CVD related to diabetes?
Are depression and CVD related to physical activity?
Are depression and CVD related to obesity?
Is it worth conducting CVD prophylaxis in patients with depression?
Is it worth conducting depression prevention in patients with CVD?
Is it advisable and necessary to conduct further research in this field?

**Table 3 biomedicines-11-01903-t003:** The influence of selected cytokines on behavior.

Cytokine	Behavior
IL-1	Has a pivotal role in the occurrence of fatigue as assessed by a decreased resistance to forced exercise on a treadmill [129,130]
IL-1β i TNF-α	Flatten the diurnal rhythm of activity by decreasing the expression of steady-state mRNAs for clock genes that control the amplitude but not the period of activity rhythms [126]
IL-10 and insulin-like growth factor I (IGF-I)	Growth factor that behaves like an anti-inflammatory cytokine in the brain, attenuates behavioral signs of sickness induced by centrally injected LPS [126]
IL-6	Chronic mild stress showed anhedonia and increased levels of circulating pro-inflammatory cytokines, including IL-6 [131,132]

IL-1—Interleukin 1, pro-inflammatory cytokine. IL-1β—Interleukin 1-β, pro-inflammatory cytokine. TNF-α—tumor necrosis factor-α, pro-inflammatory cytokine. IL-10—Interleukin 10, pro-inflammatory cytokine. IGF-I—insulin-like growth factor I. IL-6—Interleukin 6, pro-inflammatory cytokine.

**Table 4 biomedicines-11-01903-t004:** Selected cytokines and their relationships with atherosclerosis and depression.

Cytokine	Atherosclerosis	Depression
IL-1	Changes the functions of cardiac myocytes and cells in the blood vessel wall, impairing systolic function, and may intensify ischemia-reperfusion injury and expansive cardiac remodeling [138]	Can cause mood disorders, a key mediator in various behavioral effects of stress [139]
IL-6	Impaired function of vascular mitochondria accelerates the development of atherosclerosis. In mouse studies, it caused dysfunction associated with increased levels of the inflammatory cytokine IL-6 in the aorta. Human and mouse studies—aging leads to the deterioration of vascular mitochondrial function and the impairment of mitophagy. The aging of blood vessels and bone marrow cells is associated with IL-6 signaling [140]	was tested on animals and in clinical trials. Increased IL-6 activity is a factor contributing to the development of depression by activating the hypothalamic-pituitary-adrenal axis or by affecting the metabolism of neurotransmitters [141]
TNF-α	The development of atherosclerosis is closely related to the activation of TNF-α, and promotes various inflammatory reactions associated with atherosclerosis, induces vascular adhesion molecules and the recruitment and proliferation of monocytes/macrophages, participates in lipid metabolism, inhibits the activity of 7α-hydroxylase and lipoprotein lipase, and enhances the production of triglycerides in the liver [142]	Elevated plasma concentrations of tumor necrosis factor (TNF)-α in patients with mood disorders, disturbances in TNF-α levels and mental deterioration, including suicidal thoughts and response to treatment, collide [143]
IL-17	T helper-17 lymphocytes produce interleukin-17, which are important in the defense of the host mucosa against pathogenic microorganisms and fungi, constituting the anti-atherosclerotic effect of IL-17, but also the pro-atherogenic effect of IL-17 [144,145]	In depression, the number of Th17 cells increase. Th17 cells produce interleukin-17A (IL-17A), through which they promote inflammation of the nerves and activation of microglia and astrocytes. In this mechanism, they can contribute to neuronal damage, which is secondary to depression [146]

IL-1—Interleukin 1, pro-inflammatory cytokine. IL-6—Interleukin 6, pro-inflammatory cytokine. TNF-α—tumor necrosis factor-α, pro-inflammatory cytokine. IL-17—Interleukin 17, pro-inflammatory cytokine.

**Table 5 biomedicines-11-01903-t005:** Correlations between factors related to depression and CVD.

Increased risk of cardiovascular disease	Depression is associated with an increased risk of developing cardiovascular disease. Many studies show that people with depression are more likely to develop cardiovascular conditions, such as coronary artery disease, heart attacks, heart failure, and strokes.
Common Risk Factors	Depression and CVD share several common risk factors, such as a sedentary lifestyle, poor eating habits, smoking, excessive alcohol consumption, and obesity. These risk factors can contribute to both depression and cardiovascular problems.
Biological mechanisms	There are a wide range of biological mechanisms that may help explain the relationship between depression and CVD. Chronic stress, which is often associated with depression, can lead to increased inflammation and oxidative stress within the human body. These processes may contribute to the development and progression of cardiovascular diseases.
Behavioral factors	Depression can also affect an individual’s behavior and lifestyle choices in ways that increase the risk of developing CVD. For example, people with depression may engage in less physical activity, have difficulty complying with medical advice, or use unhealthy coping mechanisms, such as overeating or substance abuse.
Poor adherence to treatment	People with depression may have poor adherence to treatment required for cardiovascular disease. This lack of commitmernt can lead to inadequate treatment of risk factors, exacerbation of CVD symptoms, and increased complications.

**Table 6 biomedicines-11-01903-t006:** Results related to obesity, physical activity, and diabetes.

	Depression	Cardiovascular Disease	Depression and Cardiovascular Disease
Obesity	The prevalence of depression was 17.83% in patients with central obesity and 12.6% in non-obese patients [39]	That reducing BMI reduces the risk of CVD, at least in overweight or moderately obese patients [31]	Behavioral weight control resulted in short-term improvements in weight, mood, and CVD risk, comparable to a combination treatment of cognitive-behavioral therapy for depression [45]
	That depression and anxiety were associated with more severe obesity among adolescents seeking treatment [40]		
Physical activity	A strong antidepressant effect of exercise [52]	The impact of physical fitness and physical activity on the risk, management, and prognosis of heart failure [49]	The online cognitive-behavioral therapy group, a significant correlation was found between changes in depressive symptoms and changes in physical activity [54]
	Behavioral activation (BA) and physical activity (PA) can reduce the severity of depressive symptoms in adults [52]		Physical activity of ≥336 kcal/week had significantly lower rates of cardiovascular morbidity and mortality at 12 months [55]
	Physical exercise in a physiotherapeutic setting seems to have an effect on depressive severity and performance in major depression. Their findings suggest that physical therapy may be a viable clinical strategy that inspires and guides people with major depression to exercise [50]		Patients with coronary heart disease, exercise programs were well attended, safe, and beneficial in terms of improving physical and mental health [53]
Diabetes			The dysregulation of the hypothalamic-pituitary-adrenal HPA axis is a critical link in the high incidence of depression and comorbid diabetes [48]

**Table 7 biomedicines-11-01903-t007:** Benefits and limitations of future research.

Pros	Limitations
Improved understanding: Conducting a study can help researchers gain a better understanding of the relationship between depression and cardiovascular disease. It can shed light on the underlying mechanisms, contributing factors, and potential interactions between these two conditions.	Methodological challenges: Conducting a study on depression and cardiovascular disease can present methodological challenges. Designing appropriate research protocols, selecting appropriate study participants, and accurately measuring variables related to depression and cardiovascular health can be complex and require careful consideration.
Early detection and prevention: Identifying the link between depression and cardiovascular disease can lead to early detection and prevention strategies. This knowledge can help healthcare professionals identify individuals at risk and implement appropriate interventions to reduce the incidence of cardiovascular disease among those with depression.	Confounding factors: The relationship between depression and cardiovascular disease is multifaceted and influenced by various confounding factors. It can be challenging to isolate the effects of depression on cardiovascular health, as other factors such as lifestyle habits, socioeconomic status, and genetic predisposition may also contribute to the observed outcomes.
Treatment strategies: A study can inform the development of targeted treatment strategies. Understanding the relationship between depression and cardiovascular disease can help clinicians tailor interventions that address both conditions simultaneously, leading to better overall outcomes for patients.	Causality and directionality: Establishing a causal relationship between depression and cardiovascular disease can be difficult. It is unclear whether depression leads to cardiovascular disease, cardiovascular disease contributes to the development of depression, or if there are shared underlying factors that contribute to both conditions.
Public health implications: Findings from a study on depression and cardiovascular disease can have significant public health implications. It can inform public health policies, healthcare guidelines, and educational campaigns aimed at raising awareness about the relationship between these conditions and promoting preventive measures.	Ethical considerations: Studying depression and cardiovascular disease may involve ethical considerations. Researchers must prioritize participant welfare, ensure informed consent, and address potential psychological distress that may arise during the study. Safeguarding participant confidentiality and privacy is also crucial.

## Data Availability

Not applicable.

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
