# Peer review of "Exploring the Heart–Mind Connection: Unraveling the Shared Pathways between Depression and Cardiovascular Diseases"

_biomedicines, 2023, doi:10.3390/biomedicines11071903_

Round 1

Reviewer 1 Report

The present review by Sobolewska-Nowak and colleagues entitled ‘Common risk factors for depression and cardiovascular disease’ is a well-written and useful summary on the current status of knowledge of epidemiology and major risk factors for the coexistence of depression and cardiovascular disease.

In general, I think the idea of this review is really interesting and the authors’ fascinating observations on this timely topic may be of interest to the readers of Biomedicines. However, some comments, as well as some crucial evidence that should be included to support the author’s argumentation, needed to be addressed to improve the quality of the manuscript, its adequacy, and its readability prior to the publication in the present form. My overall judgment is to publish this paper after the authors have carefully considered my suggestions below, in particular reshaping parts of the ‘Introduction’ and ‘Methods’ sections by adding more evidence.

Please consider the following comments:

A graphical abstract that will visually summarize the main findings of the manuscript is highly recommended.

Abstract: According to the Journal’s guidelines, this section should be presented as a short summary of about 200 words maximum that objectively represents the article. It should let readers get the gist or essence of the manuscript quickly, prepare the readers to follow the detailed information, analyses, and arguments in the full paper and, most of all, it should help readers remember key points from your paper. Please, consider rewrite this paragraph following these instructions.

In general, I recommend authors to use more references to back their claims, especially in the Introduction of this review, which I believe is lacking. Thus, I recommend the authors to attempt to expand the topic of their article, as the bibliography is too concise. Nevertheless, I believe that less than 100 articles are too low for a review article. Therefore, I suggest the authors to focus their efforts on researching relevant literature: in my opinion, adding more citations will help to provide better and more accurate background to this study. 

The objectives of this study are generally clear and to the point; however, I believe that there are some ambiguous points that require clarification or refining. I think that authors here need to be explicit regarding how they operationally investigated evidence of a link between depression and cardiovascular disease, since this is the key aim of this review.

I would ask the Authors to clarify the criteria they decided to use for studies’ collection in their review: they should specify the requirements used to decide whether a study met the inclusion/exclusion criteria of the review, describe whether they included a balanced coverage of all information that is actually available, whether they have included the most recent and relevant studies and enough material to show the development and limitations in this field of interest. Finally, I believe that they should briefly present results of all statistical syntheses conducted.

Introduction: This section would benefit from a reorganization of the sub-paragraphs. As it stands, there is confusion in terms of the flow of information. I suggest to begin with a theoretical explanation of mood disorders and the role of specific brain areas in the pathophysiology of depressive disorder. In this regard, I would suggest to add more information on pathological neural substrates of depression disorder, and on structural as well as functional abnormalities of prefrontal cortex that may affect patients’ cognitive impairments (https://doi.org/10.1016/j.neubiorev.2023.105163). In my opinion, authors could further explore relationship between the molecular regulation of higher-order neural circuits and neuropathological alterations in this neuropsychiatric disorder (DOI: 10.3390/ijms24065926).

Inflammation: In my opinion, this section would benefit from a re-organization. It should include more evidence on how the role of inflammation in the genesis of depression. 

I would ask the authors to include a proper ‘Conclusions’ and ‘Limitations and future directions’ sections before the end of the manuscript, in which authors can describe some thoughtful as well as in-depth considerations in detail and report all the technical issues brought to the surface.

References: Authors should consider revising the bibliography, as there are several incorrect citations. Indeed, according to the Journal’s guidelines, they should provide the abbreviated journal name in italics, the year of publication in bold, the volume number in italics for all the references. Also, some of the references are out of date:  please cite references from the last 10 years, particularly references from the recent 5 years.

Finally, what is the take-away message from this review article? It ends rather abruptly with no summary, no suggested directions or immediate challenges to overcome, no call to action, no indications of things we should stop trying, and only brief mention of alternative perspectives. What do the authors want us to take away from this paper?

Overall, I suggest submitting your work to an English native speaker to help with some grammar mistakes that can be found in different sections of the manuscript.

I hope that, after these careful revisions, this paper can meet the Journal’s high standards for publication. 

I am available for a new round of revision of this article. 

Best regards,

Reviewer

Minor editing of English language is required.

Author Response

On behalf of the entire team, we would like to thank you for your reviews of our work. Thank you for your time and attention, thanks to which our work can become better. We have reviewed your review and made changes.

1. A graphical abstract has been made.

2. Abstract - has been rewritten.

3. The article was expanded and the number of citations increased from 61 to 102.

4. The objectives of this study have been expanded and clarified.

5. The "materials and methods" section has been thoroughly developed again. The criteria for inclusion of studies in the work have been explained.

6. Introduction - the section has been extensively expanded.

7. Inflammation Several papers have been added to the section on the relationship between depression and inflammation, but also inflammation and cardiovascular disease.

8. References - errors in the literature have been corrected. The use of older works was intended to show how long these topics have been bothering researchers and how science has developed in this direction.

9. A new discussion, results and conclusions section has been created.

10. Unfortunately, due to the short time for corrections, it was not possible to submit the work for language corrections, but we plan to do so.

Yours faithfully,

Justyna Sobolewska-Nowak

Reviewer 2 Report

The authors conducted a review to assess the common risk factors of depression and cardiovascular disease. The authors identified several risk factors, including obesity, physical activity, diabetes, and inflammation. Supportive evidence was summarized for each risk factor. The topic of this review and the evidence synthesized is essential in light of the growing burden of these disorders globally. There are some comments.

 Comments:

1.      Introduction (Line 29 on Page 1): "Civilization diseases are a serious, still growing problem." It is unclear what is the definition of civilization disease. In addition, I suggest citing references supporting the statements of the first paragraph.

2.      Introduction: Please describe this review's objectives/aim(s) at the end of the INTRODUCTION.

3.      Materials and Methods: The authors described the eligibility criteria (Line 78), the information source (Line 70), and the search strategy (Lines 71-72). However, the selection process was not clearly described. For instance, it is unclear how many investigators screened each study, whether they worked independently and compared the results, and how disagreements were resolved. In addition, please describe the date the most recent search was conducted.

4.      Materials and Methods: "All core and review articles have been researched for information related to our questions." However, the data charting (extraction) process was not clearly described. Ideally, the description of the data charting (extraction) process should at least include the following: the specific variables for which the information (data) was sought; any data extraction forms that were used; the number of investigators reviewed each published study; whether the investigators reviewed independently and compared answers or some investigators reviewed and other investigators verified; how inconsistencies or disagreements were resolved; and any processes for obtaining and confirming data from investigators.

5.      Results: Please describe the number of articles screened, assessed for eligibility, and included in the review. For excluded articles at each stage, please explain the reasons for the exclusion.

6.      Results (Depression and cardiovascular disease): This is a significant part that lays the foundation of this review. I suggest summarizing the evidence supporting the bi-directional relationship between depression and cardiovascular disease in a table.

7.      Abstract: Please describe the objectives of this review and the eligibility criteria for the review.

English language editing is recommended.

Author Response

Dear Reviewer,

On behalf of the entire team, we would like to thank you for your reviews of our work. Thank you for your time and attention, thanks to which our work can become better. We have reviewed your review and made changes.

1. Introduction (Line 29 on Page 1) - added references and sources.

2. Introduction - the section has been extensively expanded, the objectives of this study have been expanded and clarified.

3, 4. The "materials and methods" section has been thoroughly developed again. The criteria for inclusion of studies in the work have been explained.

5.6. A new discussion, results and conclusions section has been created. We created the table.

7. Abstract - has been rewritten.

Unfortunately, due to the short time for corrections, it was not possible to submit the work for linguistic corrections, but we plan to do so.

Yours faithfully,

Justyna Sobolewska-Nowak

Reviewer 3 Report

Dear Authors,

I have recently received your manuscript titled "Common risk factors for depression and cardiovascular disease" for review. The study aims to explore the relationship between depression and cardiovascular disease by discussing common risk factors such as obesity, diabetes, and lack of physical activity. While this is an important topic, I have a few concerns regarding the manuscript's novelty and structure, which I have outlined below:

Comments:

1.       The potential relationship between depression and cardiovascular disease has been extensively studied over the past twenty years. As a result, many researchers have shifted their focus from the general relationship, which is widely accepted, to specific mechanisms of predisposition, pathogenesis, and prognosis. It is essential to clearly establish the scientific novelty of this review, as the current focus appears to be well-established within the field.

2.       The structure of the review does not seem to be either chronological, which would show the development of ideas over the past years, or systematic. It is crucial to clarify the purpose of this review and organize the content accordingly.

3.       The authors' interpretation of the complex relationship between depression and cardiovascular disease appears to be somewhat loose (Figure 2). For instance, the manuscript does not seem to delve into the connection between inflammation and depression, as seen in recent studies (e.g., work by Michael Maes et al (1995-2022). It is worth noting that many cardiovascular diseases, such as atherosclerosis and diabetes, have inflammation as their main pathogenesis mechanism. Additionally, physical activity has a modulating effect on the severity of depression and inflammation.

4.       The literature search strategy should be detailed more explicitly, including the criteria for selecting articles and the reasons for excluding certain studies. Providing this information will help ensure the validity and reliability of your review.

5.       Overall, the text is well-written and uses appropriate academic language.

In light of these concerns, I recommend revising the manuscript to address the issues mentioned above. It is essential to focus on the novelty and structure of the review, provide a more accurate representation of the current understanding of the relationship between depression and cardiovascular disease, and detail the literature search strategy more explicitly.

Author Response

Dear Reviewer,

On behalf of the entire team, we would like to thank you for your reviews of our work. Thank you for your time and attention, thanks to which our work can become better. We have reviewed your review and made changes.

1. The introduction has been extensively developed.

2. The objectives of this study have been expanded and clarified. Expanded all work.

3. A new discussion, results and conclusions section has been created

4. The "materials and methods" section has been thoroughly developed again. The criteria for inclusion of studies in the work have been explained.

 Yours faithfully,

Justyna Sobolewska-Nowak

Round 2

Reviewer 1 Report

The authors did an excellent job clarifying all the questions I have raised in my previous round of review. Currently, this paper is a well-written, timely piece of research and provides a useful summary of the existing status of knowledge on the epidemiology and major risk factors for the coexistence of depression and cardiovascular disease.

Overall, this is a timely and needed work. It is well researched and nicely written, with a good balance between descriptive and narrative text.
I believe that this paper does not need a further revision, therefore the manuscript meets the Journal’s high standards for publication.

I am always available for other reviews of such interesting and important articles.
Reviewer

Author Response

Dear Reviewer,
Thank you very much for your opinion. Your comments have contributed to improving the quality of our work.
Yours faithfully,
Justyna Sobolewska-Nowak

Reviewer 3 Report

Dear Authors,

Thank you for revising and resubmitting your manuscript entitled "Common risk factors for depression and cardiovascular disease". I appreciate your efforts in addressing the comments provided in the initial review. However, upon careful examination, there are still several areas that require additional attention and clarification.

Graphic Abstract: The current version of the graphic abstract could benefit from a revision to better reflect the connection between diseases of civilization, including depression and cardiovascular diseases. The current depiction may lead to some confusion as it suggests these diseases are on opposite vectors despite sharing common developmental and risk factors. Please consider revising this figure to more accurately represent the interrelationship and commonalities between these conditions.

Figure 2: The diagram currently suggests a unidirectional influence of cytokines, potentially indicating only a peripheral response. However, cytokine response also occurs within the brain, which is not adequately represented in the figure. Consider adding a double-headed arrow or make other modifications to illustrate the bidirectional nature of cytokine response.

Materials and Methods: I noticed that this section has been slightly rephrased, but it lacks sufficient detail on the specifics of your methodology, including search query terminology and the criteria for study selection. To improve transparency and replicability, please provide a more detailed account of your methodology, possibly including a flowchart.

Revisions: It is difficult to identify the extent of the revisions made since the initial submission. I recommend that for ease of review, please highlight the changes made in the revised manuscript in color. This will help in clearly distinguishing the alterations made in response to the previous feedback.

Addressing these points will significantly improve the clarity and impact of your manuscript. I look forward to reviewing a more thoroughly revised version of your work.

Best Regards,

Author Response

Dear Reviewer,
On behalf of the entire team, thank you very much for this comment. Thanks to this, the work can become more transparent and gain in quality.
Graphic Abstract: Corrected as advised.
Figure 2: Corrected arrows.
Materials and Methods: The chapter was revised again, a flowchart was made.
Revisions: For a better overview of changes, changes are marked in green. Changes were also introduced in the graphic design of figures and tables.
Attached is a file with marked changes.

Yours faithfully,
Justyna Sobolewska-Nowak

Round 3

Reviewer 3 Report

Dear Authors,

 Re: Manuscript Title: Common Risk Factors for Depression and Cardiovascular Disease

 I have had the pleasure of reviewing your manuscript and I would like to express my appreciation for the work you have undertaken. The manuscript has significantly improved in quality and provides a comprehensive and insightful exploration of the common risk factors for depression and cardiovascular disease.

 In principle, I am supportive of the acceptance of your paper by the journal in its current form. However, I would like to suggest the incorporation of some additional references that could further strengthen the sections on atherosclerosis, systemic inflammation, and the role of physical activity.

 In the section discussing atherosclerosis and systemic inflammation (lines 337-361), I recommend the review by Gusev and Sarapultsev entitled "Atherosclerosis and Inflammation: Insights from the Theory of General Pathological Processes" (Int. J. Mol. Sci. 2023, 24, 7910. https://doi.org/10.3390/ijms24097910). This paper provides an excellent overview of the intricate relationship between atherosclerosis and inflammation and could provide additional depth to your discussion.

 Furthermore, in the section discussing inflammation and physical activity (lines 238-278), the review by Calcaterra et al., "Use of Physical Activity and Exercise to Reduce Inflammation in Children and Adolescents with Obesity" (Int. J. Environ. Res. Public Health 2022, 19, 6908. https://doi.org/10.3390/ijerph19116908) might provide useful insights. This paper discusses the beneficial effects of physical activity in reducing inflammation, which is a pertinent point that aligns well with the theme of your manuscript.

 I believe that the addition of these resources could further enrich your discussion and provide readers with a more comprehensive understanding of the subject matter. Once again, I would like to commend your team for the excellent work on this manuscript.

 Looking forward to seeing this contribution to the field published.

 Kind regards,

Author Response

Dear Reviewer,
Thank you very much for your valuable comments, which helped to improve the quality of our work.
Thank you for pointing out the interesting articles we have included in our review.
Attached is the file with the changes marked in green.
Yours faithfully,
Justyna Sobolewska-Nowak